# Exploring a Paradigm Shift in Primary Teeth Root Canal Preparation: An Ex Vivo Micro-CT Study

**DOI:** 10.3390/children10050792

**Published:** 2023-04-28

**Authors:** Dora Schachter, Sigalit Blumer, Sara Sarsur, Benjamin Peretz, Tatiana Sella Tunis, Shada Fadela, Johnny Kharouba, Shlomo Elbahary

**Affiliations:** 1Department of Pediatric Dentistry, The Maurice and Gabriela Goldschleger School of Dental Medicine, Tel Aviv University, Ramat Aviv, Tel Aviv 6997801, Israel; 2Department of Orthodontics, The Maurice and Gabriela Goldschleger School of Dental Medicine, Tel Aviv University, Ramat Aviv, Tel Aviv 6997801, Israel; 3Department of Endodontology, The Maurice and Gabriela Goldschleger School of Dental Medicine, Tel Aviv University, Ramat Aviv, Tel Aviv 6997801, Israel

**Keywords:** pulpectomy, micro-CT, Kedo S, protaper gold, manual preparation, rotary instrumentation, primary teeth, mandibular molars

## Abstract

Traditional hand instrumentation is a commonly used technique to perform pulpectomy in deciduous teeth by most specialists in pedodontics. Lately, dentists have embraced rotary instrumentation as a safe and effective alternative. This study aimed to compare the efficacy of root canal preparation in extracted primary molars between manual and two rotary file systems using micro-CT. Thirty-six extracted human second primary mandibular molars were divided into three groups according to the instrumentation method: (1) Manually instrumented (MI) group (n = 9) in which the teeth were treated using K-files up to size 30; (2) Kedo-Ssystem (KS) group (n = 9); (3) ProTaper Gold system (PTG) group (n = 10) and control group (n-8). Each tooth was scanned before and after the retrograde root canal preparation. Residual dentin volume was calculated using micro-CT scans to evaluate the technique’s efficacy. Additionally, the preparation time and procedural errors were recorded for each tooth preparation. A one-way ANOVA test was carried out to compare the groups’ dentin volume and preparation time. The mean preparation time using the manual method (13.14 min) was more than two times longer than that of the rotary techniques (4.62 min and 6.45 min). The manual preparation method using a K-file removed almost half the root canal material when compared with the rotor method (*p* = 0.025). Conclusion: our results suggest that rotary instrumentation is more efficient for root canal preparation in primary teeth than the traditional manual method. This finding may call for a paradigm shift in current clinical practices, where manual instrumentation is still commonly preferred.

## 1. Introduction

A pulpectomy is a conservative treatment method used to prevent premature loss of primary teeth with irreversible pulpitis or necrotic pulp, which can cause a loss of arch length, insufficient space for erupting permanent teeth, impaction of premolars, and tipping of the teeth adjacent to the lost primary teeth [1,2]. The success of the pulpectomy procedure mainly depends on the chemo-mechanical preparation of the root canal system [3]. Manual instrumentation during pulpectomy in primary teeth is usually done using filing, but this system has limitations regarding patient management, chemo-mechanical circumferential root canal preparation, and time consumption [3]. An efficient pulpectomy must be quick, simple, and effective in disinfection without compromising tooth structure or the permanent tooth bud, preventing procedural errors, and restoring the tooth’s function and integrity [4,5].

As mentioned earlier, chemo-mechanical preparation is crucial for the success of pulpectomy [1]. Thorough cleansing and shaping of the root canal system help extirpate the infected pulp tissue, provide straight-line access for irrigation solutions to reach the apical part of the root, and allow for adequate filling of the root canals [6,7]. In primary teeth, it is challenging to accomplish sufficient preparation, mainly due to the complex root canal anatomy, which is extremely rich in ramifications, anastomosis, accessory and secondary canals [8,9].

Rotary instrumentation has been used in endodontics for decades to prepare the root canal, revolutionizing endodontic treatment by making it faster, more efficient, and more predictable [3,10,11]. Rotary instruments are made of a nickel–titanium (NiTi) alloy, which has several advantages over traditional stainless-steel (SS) files [11]. NiTi files are more flexible and have excellent resistance to fracture, reducing the risk of instrument separation during use. They also have an exceptional cutting efficiency due to their shape memory and super elasticity qualities, allowing for an efficient and safe shaping of the root canal system [12].

In 2000, Barr et al. [13] introduced the use of rotary instrumentation in primary teeth and its advantages in performing adequate preparation for the efficient cleaning and shaping of the canal. The use of rotary files in primary dentition decreased preparation time. NiTi rotary instruments also provide a suitable taper for rinsing and obturation with minimal risk of misshapes [14,15]. Nevertheless, using rotary files also has drawbacks, such as high cost, file separation, or perforation [16].

The Kedo-S pediatric rotary instrumentation system (Reeganz Dental Care Private Limited, Chennai, India) was designed exclusively for primary teeth. It is a newer system specifically intended to make root canal treatment in children faster and more efficient. The files have a unique design that allows them to follow the curvature of the root canal, reducing the risk of instrument fracture and improving the procedure efficiency [17,18]. The system is designed to be easy to use, even for dentists who may not have extensive experience with rotary endodontic techniques. The Kedo-S files have a total length of 16 mm and a working length of 12 mm with a gradual taper [19]. This creates an advantage for use on primary teeth, which have shorter, thinner, curved roots and a ribbon-shaped morphology compared with permanent teeth [2].

Various methods have been used in the past to evaluate the efficacy of root canal rotary instrumentation, including radiographs, histological sections and scanning electron microscopy (SEM). However, each of these methods has limitations: radiographs are two-dimensional and may not accurately reflect the three-dimensional complexity of the root canal system, while histological sections are time-consuming and can alter the morphology of the sample. SEM is limited to surface morphology and requires the use of a vacuum that can alter the sample [9,20,21].

Micro-computed tomography (micro-CT) is a non-destructive imaging technique that allows for high-resolution 3D imaging of small structures, including primary teeth. micro-CT can be used to visualize and quantify the internal structure of primary teeth, including the root canal system and dentin thickness, which can be helpful for diagnostic and treatment planning. Moreover, micro-CT can assess the quality and accuracy of root canal preparation and filling in primary teeth. It can provide information about the shape and volume of the root canal system, the presence of voids or gaps in the filling material, and the thickness and integrity of the root canal walls. This can ensure optimal treatment outcomes and reduce the risk of complications, such as persistent infection or root fracture. It is also used to evaluate uninstrumented canal surfaces and shape the efficacy of rotary and manual file systems [9,20,21].

Few studies used micro-CT to compare manual instrumentation and rotary files in primary tooth preparation. However, to the best of our knowledge, none of these studies present a validation of the suggested model.

Therefore, this study aimed to compare the efficacy of root canal preparation in human-extracted primary molars between a manual and exclusively designed Kedo-S rotary files system, and ProTaper gold, using micro-CT.

## 2. Materials and Methods

### 2.1. Experimental Teeth

Thirty-six sound-extracted human second primary mandibular molars, with no physiological root resorption, were collected and stored in distilled water. Teeth with pathological resorption, fractures, severe curve (a root canal curvature > 20° [22]), perforation in the furcation area, and less than two thirds of the root were excluded from the study.

Access cavities were prepared using 330 high-speed carbide bur (Mailleffer, Ballaigues, Switzerland) with water spray. A size 10 K-file (Dentsply Maillefer, Tulsa, OK, USA) was placed into the canals until it was visible at the apical foramen, and the working length was established as 1 mm short of a length. All specimens were mounted vertically in auto-polymerizing acrylic resin (UNIFAST Trad, Gc America).

### 2.2. Tooth Instrumentation

To avoid bias errors, all the pulpectomies were prepared by a single operator (S.E).

The teeth were divided into the following groups:

(1)MI (manual instrumentation) group (n = 9): For the root canal instrumentation procedure, the operator used the balanced force technique [23] with K-files (Dentsply Maillefer, Tulsa, OK, USA). Initially, a size 10 K-file was inserted into the canal, and the working length determined and confirmed by a radiograph. The file was rotated in a clockwise and counterclockwise motion with light pressure to create a glide path for larger files. The canal was then recapitulated with smaller hand files to remove debris and verify the working length. The operator then selected a size-15 K-file and repeated the process of preparation and recapitulation. This process was repeated with progressively larger K-files up to size 30, until the desired apical size and taper were achieved. Recapitulation with smaller hand files and irrigation with a sodium hypochlorite solution were performed throughout the procedure. The operator used balanced force and controlled movements to minimize the risk of iatrogenic errors, such as canal transportation or perforation.(2)KS (Kedo S) group (n = 9): The teeth were instrumented with a Kedo-S system (16 mm) (Reeganz Dental Care Private Limited, Chennai, India). The system included the usage of three files: (a) white (17/08), (b) yellow (20/04), and (c) red (25/04). For the root canal instrumentation procedure, the operator used a Kedo-S rotary file system (Reeganz Dental Care Private Limited, Chennai, India). After establishing access to the root canal system and determining the working length, the operator inserted the appropriate size Kedo-S file to the working length and used a gentle filing motion in clockwise and counterclockwise directions to create a glide path. Recapitulation with smaller hand files was performed to remove debris and verify the working length. This process was repeated with progressively larger Kedo-S files until the desired apical size and taper were achieved. Throughout the procedure, irrigation with sodium hypochlorite solution was used to clean the canal and remove debris. The operator used controlled and gentle movements to minimize the risk of iatrogenic errors, such as canal transportation or perforation.(3)PTG (ProTaper Gold) group (n = 10): For the root canal instrumentation procedure, the operator used a ProTaper Gold rotary file system (Dentsply Maillefer, Ballaigues, Switzerland). After establishing access to the root canal system and determining the working length, the operator used a size 10 hand file to establish a glide path. The S1 file was then used to shape the coronal portion of the canal. The SX file was used to finish shaping the canal or to bypass ledges or obstructions. The operator then used the S1, S2, and F1 files in a sequential manner to shape the middle and apical portions of the canal. Recapitulation with smaller hand files was performed to remove debris and verify the working length. The operator then used the F2 file to achieve the desired apical size and taper. Throughout the procedure, irrigation with a sodium hypochlorite solution was used to clean the canal and remove debris. The operator used controlled and gentle movements to minimize the risk of iatrogenic errors, such as canal transportation or perforation.(4)Control (n = 8) group: no instrumentation and irrigation were performed in the root canals. The control group aimed to evaluate the reliability of dentin volume evaluation using micro-CT.

After the preparation, the teeth from all the treated groups were irrigated with 4 mL of saline and dried with absorbent paper points.

### 2.3. Micro-CT Scanning

For micro-CT scanning of the teeth, the operator used the Vector4CT scanner (Bruker Micro-CT, Kontich, Belgium) system for all specimens two times (before and after the preparation). The teeth were first isolated from the surrounding tissue and fixed in a container with a radiopaque medium. The container was then placed in the scanner and the scanning process was initiated. The exposure parameters were 50 kV, 0.43 mA, isotropic resolution of 21 μm, and a 360° rotation. After completion of the scanning process, the images were reconstructed using Milabs Reconstruction 10.16 SN80843 software. Voxel size of 20 µm × 20 µm × 20 µm. The root canal volume was evaluated using the DragonFly Software version 2022.2 (Objects Research Systems, Montreal, QC, Canada) (Figure 1). Throughout the process, the operator followed the manufacturer’s instructions for use and maintained the necessary safety precautions.

The original grayscale images were processed with Gaussian low-pass filtration for noise reduction and a fixed segmentation threshold to separate root dentin from the root canal.

The region of interest was selected, extending from mesial and distal canals separated at the furcation area to the apex. To ensure the part of interest at the second scanning, the distance from CEJ was measured. The mesial and distal root canal volume was calculated separately (Figure 2).

### 2.4. Preparation Time and Errors

Root canal preparation time (sec): the time required to prepare the canals. The time was recorded using a stopwatch and measured from the beginning of the instrument activation in the root canal until the end.

Procedural errors: every procedural error, such as instruments separation, perforation, and deviation, were recorded. Additionally, a blind evaluation of micro-CT images was performed by an experienced evaluator to obtain information about parameters such as the presence/absence of procedural errors, including instrument separation, root canal perforation, and deviation. Each sample scan was evaluated before the root end preparation to exclude procedural errors that could appear due to canal preparation. After confirming that no procedural errors were found in the first scan, we proceeded to the second scan.

### 2.5. Statistical Analysis

The SPSS software package (Statistical Package for Social Sciences, version 20.0, SPSS Inc., Chicago, IL, USA) was used to record and analyze the data in this study. Normal distribution of all measurements was confirmed using a one-sample Kolmogorov–Smirnov test. To identify significant differences in root canal volume and preparation time among the different preparation methods, a one-way ANOVA test was performed. Post hoc multiple comparisons were then conducted to determine significant differences between the groups. Statistical significance was defined as *p* < 0.05.

### 2.6. Reliability

To determine the ability to accurately replicate the measurement of the root canal volume using micro-CT, eight different primary second molar teeth (a total of 16 root canals) were scanned and evaluated twice, with a two-week interval.

## 3. Results

### 3.1. Reliability Analysis

Excellent reliability results (ICC = 0.995, *p* < 0.0001) were found when evaluating the ability to reproduce the volume of the root canal in the primary second molars.

### 3.2. Root Canal Preparation Time

A significant difference was found in the preparation time between manual and rotor methods (*p* < 0.0001) (Table 1, Figure 3). The mean preparation time using the manual mode (13.14 min) was more than two times longer than that of the rotor techniques (4.62 min and 6.45 min). Nevertheless, no statistically significant difference was observed in preparation time between the rotor preparation methods (*p* = 0.064) (Figure 3).

### 3.3. Root Canal Volume

Root canal volume was evaluated twice before the root canal preparation (“initial” evaluation period) and following root canal preparation (“final” evaluation period). The increase in the root canal volume was calculated by subtracting the initial volume from the final volume (“change”), and descriptive statistics for each preparation technique are presented in Table 2.

No significant difference was found in the initial root canal volume before the root canal preparation using the three instruments (*p* = 0.052). Our results showed that the manual preparation method using K-file removes almost half the root canal material when compared with the rotor method (*p* = 0.025) (Table 2, Figure 4). No significant difference was found in the root canal volume change between the two rotor preparation methods (*p* = 0.803) (Table 2, Figure 4).

## 4. Discussion

In this study, we presented a novel, innovative, and efficient measurement method based on a micro-CT device (VECTor4/CT type micro-CT instrument; MILabs, Utrecht, The Netherlands) for the three-dimensional assessment of root canal dimensions during RCT in primary teeth. Although several studies have evaluated changes in preparation in the past, the level of reliability of those studies remained unclear, as no reliability tests were noted in those manuscripts. Compared to those studies, in the present study, the reliability of the volume measurements of root canals in primary second molars was evaluated using intra-class correlation coefficient (ICC) analysis. The results revealed excellent reliability, with an ICC value of 0.995, indicating a high degree of agreement between the repeated measurements. This suggests that the method used for measuring root canal volumes is highly reproducible and can be considered reliable for use in future studies. Additionally, the *p*-value of less than 0.0001 indicates that the observed agreement between the measurements is highly statistically significant, further supporting the reliability of the method. These findings have important implications for the use of volume measurements as an outcome measure in studies investigating the effectiveness of different endodontic treatment techniques in primary teeth. Overall, the excellent reliability of the volume measurements obtained in this study provides a foundation for future investigations aimed at improving the quality of endodontic treatment in primary teeth.

Although the conventional hand file technique has been used for years and is considered the gold standard in pediatric dentistry, it is time-consuming and plays a crucial role in behavior management in pediatric dentistry. In this study, we used two NiTi systems, the ProTaper Gold and the new Kedo-S files, and found that both rotary systems resulted in a significantly lower preparation time (4.62 min and 6.45 min) than manual preparation (13.14 min). We found no statistically significant difference between rotary preparation methods (*p* = 0.064). These data are consistent with previous investigations [9,10,24,25]. In 2014, Musale et al. [10] found that the mean time for manual preparation using K-files was 20.8 min, while using the ProTaper Universal was 5.8 min. ProTaper Universal and ProTaper Gold systems have a matching instrument object with a triangular cross-section and a variable progressive taper. PTG is manufactured by proprietary metallurgy that reportedly increases its flexibility and resistance to cyclic fatigue, which may explain the lower preparation time using PTG [26].

Primary teeth have a softer root dentin, and short, thin, curved, and ribbon-shaped root morphology [27]. While cleaning and shaping a primary tooth, maintaining the remaining dentin thickness is essential, as there is a direct correlation between root thickness and fracture resistance. Lim and Stock [28] have suggested that a minimum of 0.3 mm thickness of canal walls should remain after canal preparation, which allows for adequate resistance against lateral and occlusal forces. As the amount of dentin removal indicates the aggressiveness of the instrument, we evaluated the volume of the removed dentin in this study. The present study found that rotary instrumentation resulted in a double canal volume in half the time. This difference can be explained by the higher taper of both rotary systems compared to the 0.02 taper of the K-file. Previous in vitro studies [9,10,24,26,29] that compared dentin removal using the rotary and manual methods showed no significant difference between the methods or that removal was more conservative in the rotary method compared to the manual method. However, in our study, we found the opposite result. We found that with the rotary methods (combined), there was a significantly greater dentin removal of the root canal than with the manual method. The cited studies are in vitro studies that tested manual or rotary preparation under ideal conditions and indicated a manual preparation time of approximately 20 min, while with the rotary method, about 5–6 min. However, under in vitro conditions, a 20 min preparation during RCT in primary teeth cannot stand in a clinical situation due to the need for the child’s cooperation. A clinical comparison study [30] of Kedo-S pediatric rotary files with manual instrumentation during root canal preparation in primary molars found that the manual preparation time was 95.46 s compared to 78.53 s in the rotary method. Therefore, the argument presented in the discussions that the manual method removes a more significant amount of dentin due to a longer working time is invalid in a clinical setting. In permanent dentition, it is widely acknowledged that the use of rotary instrumentation results in a greater dentin removal of the root canal compared to the traditional manual method [31,32]. This is due to the design and shape of rotary instruments, which allows for more efficient and faster removal of dentin. However, the efficacy and safety of rotary instrumentation in primary teeth have been a matter of debate among pediatric dentists. This study aimed to compare the efficacy of root canal preparation in primary molars between manual and two rotary file systems, using micro-CT. The results showed that the use of rotary instrumentation in primary teeth resulted in a significantly greater dentin removal of the root canal compared to the traditional manual method, which is consistent with previous research in permanent dentition.

Efficient root canal instrumentation is crucial to remove infected content and create a root canal shape that allows for a well-condensed root filling. From a clinical standpoint, effective canal preparation is necessary for successful treatment outcomes and is more feasible with rotary instrumentation. Similar to other studies, we found no significant differences in preparation time or canal volume between the two rotary systems. The Kedo-S file system was introduced [19] to modify the current rotary instrumentation. It consists of three NiTi files (D1, E1, U1), with an altered working length of 12 mm, to expedite its use only in primary teeth. Another added feature of these files is the presence of a variable taper. D1 and E1 files have been designed for the instrumentation of molars, with a tip diameter of 0.25 and 0.30 mm, respectively. The D1 file has 4, 5, 6, and 8% tapers at different lengths, allowing the file to be used in narrower canals in primary molars, primarily the mesiobuccal and mesiolingual canals. The E1 file has 4, 6, and 8% tapers at different lengths, corresponding to the wider canals in primary molars, mainly the distal canals. While rotary instrumentation has become more popular for endodontic treatment in permanent teeth, its use in primary dentition is still somewhat controversial. Many pediatric dentists continue to prefer manual instrumentation for pulpectomy in primary teeth due to the unique anatomy and morphology of primary teeth or their lack of experience. Although some studies have suggested that rotary instrumentation can be effective and safe for primary teeth, the choice of instrumentation technique ultimately depends on the specific case and the skill and experience of the dentist. Both manual and rotary instrumentation have advantages and limitations, and the decision should be made on a case-by-case basis to ensure the best possible outcome for the patient.

One of the limitations of this in vitro study is that it cannot replicate the dynamic environment of the oral cavity and was conducted in controlled laboratory settings that do not reflect the true complex conditions. We acknowledge that the small number of teeth in each group is a limitation of our study. However, it is important to note that an ex vivo study with micro-CT analysis is a time-consuming and resource-intensive process, which limits the number of teeth that can be included. Moreover, in order to obtain reliable and accurate results, it is necessary to ensure that the sample size is adequate and that the teeth are selected and prepared consistently. We chose to include nine teeth in each of the manual and Kedo-S groups, and 10 teeth in the ProTaper Gold group, based on previous studies that used similar methodologies. While a larger sample size would have provided more statistical power and increased the generalizability of our findings, we believe that our results still provide valuable insights into the efficacy of different root canal preparation methods in primary teeth.

## 5. Conclusions

Although this study was limited to an ex vivo setting, our results suggest that rotary instrumentation is more efficient for root canal preparation in primary teeth compared to the traditional manual method. This finding may call for a paradigm shift in current clinical practices, where manual instrumentation is still commonly preferred. Further studies in clinical settings are needed to confirm our findings and to evaluate the long-term outcomes of using rotary instrumentation in primary teeth.

## Figures and Tables

**Figure 1 children-10-00792-f001:**
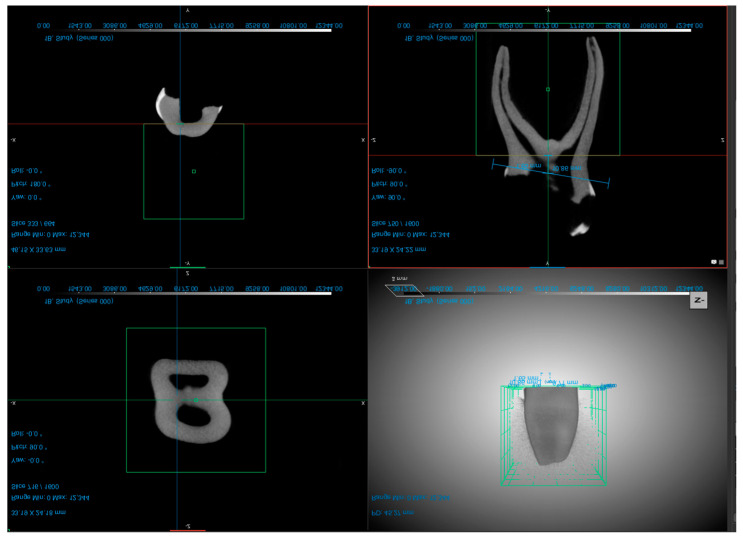
Examples of different planes of the second primary molar (screenshots taken from the DragonFly Software version 2022.2.

**Figure 2 children-10-00792-f002:**
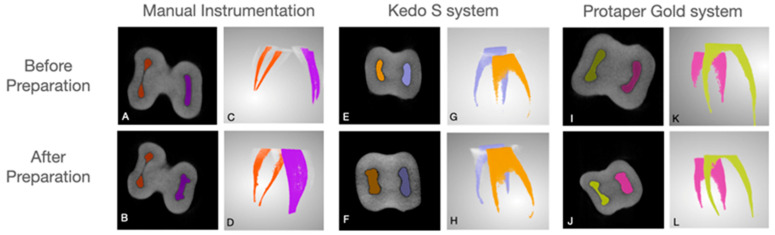
Examples of second primary molars instrumented with the different systems viewed on micro-CT scans: manual instrumentation before (**A**,**C**) and after (**B**,**D**) preparation, Kedo S system before (**E**,**G**) and after preparation, (**F**,**H**) and Protaper Gold before (**I**,**K**) and after preparation (**J**,**L**).

**Figure 3 children-10-00792-f003:**
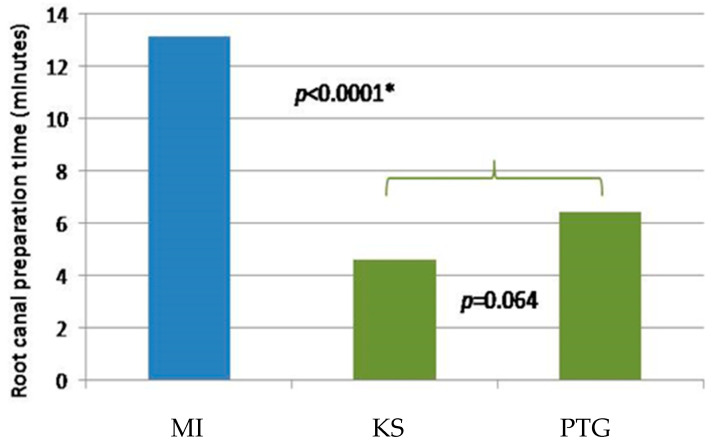
Graphic presentation of the mean preparation time (in min) of different methods. The blue bar represents the manual technique using a K-file, green bars represent rotor preparation techniques (MI, KS, PTG). * MI, statistical significance different.

**Figure 4 children-10-00792-f004:**
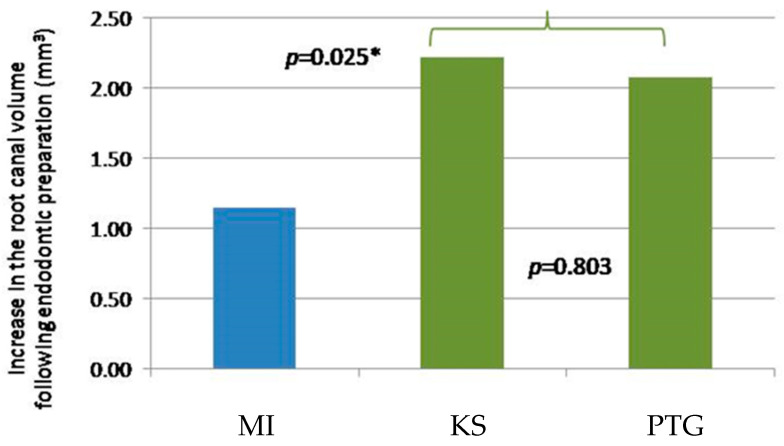
Graphic presentation of the increase in the root canal volume following endodontic preparation using different techniques (blue bar represents the MI technique using a K-file, green bars represent rotor preparation techniques (KS and PTG)). * MI, statistical significance different.

**Table 1 children-10-00792-t001:** Descriptive statistics of the preparation time (seconds) of different methods.

Preparation Method	Instrument Type	n (Teeth)	Mean (s)	±SD	Min.	Max.	*p*-Value *
Manual	MI	9	788.11	127.486	660	971	**<0.0001**
Rotor	KS	9	277.11	67.953	180	418	
	PTG	10	387.10	85.239	243	543	

* Significant *p*-values are denoted in bold. SD: standard deviation; Min: minimum; Max: maximum.

**Table 2 children-10-00792-t002:** Descriptive statistics of the root canal volume measured before and after the treatment using different preparation methods.

Preparation Method	Instrument Type	n (Canals)	Evaluation	Mean (mm^3^)	±SD	Min.	Max.
			Period				
Manual	MI	18	Initial	8.13	3.607	3.59	17.85
			Final	9.28	4.118	3.9	19.33
			Change	1.15	1.119	0.47	3.66
Rotor	KS	18	Initial	6.92	4.958	1.18	18.36
			Final	9.13	6.123	1.6	27.37
			Change	2.22	2.134	0.14	9.01
	PTG	20	Initial	5.03	2.715	1.28	11.89
			Final	7.11	3.214	3.15	13.14
			Change	2.08	1.110	0.65	4.64

## Data Availability

The data supporting the results are available from the corresponding authors upon reasonable request.

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
