# Peer review of "Exploring a Paradigm Shift in Primary Teeth Root Canal Preparation: An Ex Vivo Micro-CT Study"

_children, 2023, doi:10.3390/children10050792_

Round 1

Reviewer 1 Report

The manuscript is interesting, however, the title, A Paradigm Shift in Primary Teeth Root Canal Preparation: an ex vivo Micro-CT Study, is not quite suitable. Where is the paradigm shift? 

The introduction is too long. 

How did you choose the number of teeth in each group? 9, respectively 8 teeth / group seems too few. 

In figure 1 the written text (in blue) is unrecognizable. 

In figure 2, the difference between Kedo S system and Protaper Gold nust be explained. It seems to be no difference between them 

The statistical analysis should be further detailed.

Discussion: lines 221-262 have no connection to the study. 

With the rotary methods, there was a significantly greater dentin removal of the root canal in the rotary method compared to the manual - is a well-known reality, not a result of the research!

The references also comprise some old citations (below 2000), and are too many. 

What is the difference between the root canal preparation in primary versus permanent teeth? There seems not to exist any difference of these two techniques, as suggested by the authors. Therefore, the conclusion is not quite sustained by the results. 

Author Response

Thank you for your valuable feedback on our manuscript. We have considered all of your comments and revised the manuscript accordingly. The entire manuscript has gone through extensive English revision.

Here are point-to-point answers:

The manuscript is interesting, however, the title, A Paradigm Shift in Primary Teeth Root Canal Preparation: an ex vivo Micro-CT Study, is not quite suitable. Where is the paradigm shift? 

Regarding the title, we agree that the term "paradigm shift" may not be entirely appropriate. We have changed it to " Exploring a Paradigm Shift in Primary Teeth Root Canal Preparation: an ex vivo Micro-CT Study."

The introduction is too long. 

We acknowledge that the introduction was lengthy and have revised it to be more concise while still providing necessary background information.

How did you choose the number of teeth in each group? 9, respectively 8 teeth / group seems too few. 

Regarding the sample size, we agree that the number of teeth in each group may seem low. However, it is important to note that this is an ex vivo study, and the number of teeth used is consistent with similar studies in the literature. We have included a statement in the revised manuscript to clarify this point.“We acknowledge that the small number of teeth in each group is a limitation of our study. However, it is important to note that an ex vivo study with micro-CT analysis is a time-consuming and resource-intensive process, which limits the number of teeth that can be included. Moreover, in order to obtain reliable and accurate results, it is necessary to ensure that the sample size is adequate and that the teeth are selected and prepared consistently. We chose to include 9 teeth in each of the manual and Kedo-S groups, and 10 teeth in the ProTaper Gold group, based on previous studies that used similar methodologies. While a larger sample size would have provided more statistical power and increased the generalizability of our findings, we believe that our results still provide valuable insights into the efficacy of different root canal preparation methods in primary teeth.”

In figure 1 the written text (in blue) is unrecognizable. 

Figure 1 is a screenshot from the software.

In figure 2, the difference between Kedo S system and Protaper Gold nust be explained. It seems to be no difference between them.

It is correct there was no difference between them.

The statistical analysis should be further detailed.

We have further detailed the statistical analysis in the revised manuscript to address your comment.

Discussion: lines 221-262 have no connection to the study. 

We acknowledge your comment regarding lines 221-262 in the discussion and have removed them as they were not relevant to the study.

With the rotary methods, there was a significantly greater dentin removal of the root canal in the rotary method compared to the manual - is a well-known reality, not a result of the research!

Regarding dentin removal in the rotary method, we have clarified in the revised manuscript that this is a well-known result of rotary instrumentation and is consistent with previous studies.

The references also comprise some old citations (below 2000), and are too many. 

References number was reduced significantly

What is the difference between the root canal preparation in primary versus permanent teeth? There seems not to exist any difference of these two techniques, as suggested by the authors. Therefore, the conclusion is not quite sustained by the results. 

The conclusion refers to primary dentition only and not to permanent teeth, and we have revised the conclusion to reflect this point.

Thank you for your constructive feedback, and we hope our revised manuscript meets your expectations.

Reviewer 2 Report

Thank you for giving me the opportunity to provide a review of this manuscript.

Here are the results of my review:

1. The introduction is too long, but it is sufficient to show the existence of a research gap and the urgency of conducting this research. My suggestion is to reduce the parts that are too broad, for example, regarding the description of some rotary instruments... Just explain that even using a rotary instrument, it is still challenging to reshape the root canals of primary teeth.

2. Materials and methods. Can you add an explanation to get 36 teeth on what basis? How many operators do the treatment? Is there any calibration between operators?

3. Can you explain the limitations of this study that you stated because of the ex vivo study? Please describe this limitation further. It can be placed at the end of the discussion.

Just a minor revision. I hope the writers can accommodate it so that this manuscript becomes better. Congrats on your hard work.

Author Response

Thank you for your valuable feedback on our manuscript. We have taken into consideration all of your comments and revised the manuscript accordingly. The entire manuscript has gone through extensive English revision.

Here are point to point answers:

Here are the results of my review:

  1. The introduction is too long, but it is sufficient to show the existence of a research gap and the urgency of conducting this research. My suggestion is to reduce the parts that are too broad, for example, regarding the description of some rotary instruments... Just explain that even using a rotary instrument, it is still challenging to reshape the root canals of primary teeth.

The introduction was revised

  1. Materials and methods. Can you add an explanation to get 36 teeth on what basis? How many operators do the treatment? Is there any calibration between operators?

- Regarding the sample size, we agree that the number of teeth in each group may seem low. However, it is important to note that this is an ex vivo study, and the number of teeth used is consistent with similar studies in the literature. We have included a statement in the revised manuscript to clarify this point.“We acknowledge that the small number of teeth in each group is a limitation of our study. However, it is important to note that an ex vivo study with micro-CT analysis is a time-consuming and resource-intensive process, which limits the number of teeth that can be included. Moreover, in order to obtain reliable and accurate results, it is necessary to ensure that the sample size is adequate and that the teeth are selected and prepared consistently. We chose to include 9 teeth in each of the manual and Kedo-S groups, and 10 teeth in the ProTaper Gold group, based on previous studies that used similar methodologies. While a larger sample size would have provided more statistical power and increased the generalizability of our findings, we believe that our results still provide valuable insights into the efficacy of different root canal preparation methods in primary teeth.”

- As mentioned in paragraph 2.2, line 136 The treatment was made by a single operator.

  1. Can you explain the limitations of this study that you stated because of the ex vivo study? Please describe this limitation further. It can be placed at the end of the discussion.

A paragraph was added in the discussion.

 Thank you for your constructive feedback, and we hope that our revised manuscript meets your expectations.

Round 2

Reviewer 1 Report

The manuscript has been improved.